# Rbm20^ΔRRM^ Mice, Expressing a Titin Isoform with Lower Stiffness, Are Protected from Mechanical Ventilation-Induced Diaphragm Weakness

**DOI:** 10.3390/ijms232415689

**Published:** 2022-12-10

**Authors:** Marloes van den Berg, Eva L. Peters, Robbert J. van der Pijl, Shengyi Shen, Leo M. A. Heunks, Henk L. Granzier, Coen A. C. Ottenheijm

**Affiliations:** 1Department of Cellular and Molecular Medicine, University of Arizona, Tucson, AZ 85724, USA; 2Department of Physiology, Amsterdam UMC, 1082 HZ Amsterdam, The Netherlands; 3Intensive Care Medicine, Erasmus MC, 3015 GD Rotterdam, The Netherlands

**Keywords:** diaphragm weakness, mechanical ventilation, titin, sarcomere dysfunction

## Abstract

Diaphragm weakness frequently develops in mechanically ventilated critically ill patients and is associated with increased morbidity, including ventilator weaning failure, mortality, and health care costs. The mechanisms underlying diaphragm weakness are incompletely understood but may include the elastic properties of titin, a giant protein whose layout in the muscle’s sarcomeres makes it an ideal candidate to sense ventilation-induced diaphragm unloading, resulting in downstream signaling through titin-binding proteins. In the current study, we investigated whether modulating titin stiffness affects the development of diaphragm weakness during mechanical ventilation. To this end, we ventilated genetically engineered mice with reduced titin stiffness (Rbm20^ΔRRM^), and robust (Ttn^ΔIAjxn^) or severely (Ttn^Δ112–158^) increased titin stiffness for 8 h, and assessed diaphragm contractility and protein expression of titin-binding proteins. Mechanical ventilation reduced the maximum active tension of the diaphragm in WT, Ttn^ΔIAjxn^ and Ttn^Δ112–158^ mice. However, in Rbm20^ΔRRM^ mice maximum active tension was preserved after ventilation. Analyses of titin binding proteins suggest that muscle ankyrin repeat proteins (MARPs) 1 and 2 may play a role in the adaptation of the diaphragm to mechanical ventilation, and the preservation of diaphragm contractility in Rbm20^ΔRRM^ mice. Thus, Rbm20^ΔRRM^ mice, expressing titin isoforms with lower stiffness, are protected from mechanical ventilation-induced diaphragm weakness, suggesting that titin elasticity may modulate the diaphragm’s response to unloading during mechanical ventilation.

## 1. Introduction

Diaphragm weakness is often present in mechanically ventilated critically ill patients. It develops rapidly, with 64% of patients presenting with diaphragm weakness within one day after onset of mechanical ventilation and is still present in 63% of patients at day 3 [1]. At the time of ventilator weaning, 80% of patients developed diaphragm weakness [2], associated with weaning failure and mortality [3,4,5]. Thus, although initially lifesaving, mechanical ventilation is associated with significant side effects, in particular respiratory muscle weakness.

The mechanisms underlying diaphragm weakness in mechanically ventilated critically ill patients are incompletely understood, but the reduced pressure-generating capacity, thickness, and thickening fraction of the diaphragm indicate contractile dysfunction and atrophy [6,7,8,9,10]. At a cellular level, diaphragm fibers of mechanically ventilated critically ill patients also show atrophy and contractile dysfunction [11,12,13]. However, contractile dysfunction is not fully explained by atrophy because force normalized to the cross-sectional area of the muscle fibers (i.e., tension) is also lower after mechanical ventilation [11,12,13]. Similar results were found in diaphragm fibers of brain-dead organ donors [14,15,16] and mechanically ventilated animal models [9,17,18,19,20,21].

The mechanisms that drive diaphragm fiber atrophy and contractile dysfunction upon unloading are unclear but may involve the giant sarcomeric protein titin (3–4 MDa, depending on species and isoform). Titin filaments span from the sarcomeric Z-disc to the M-line, where they interconnect with adjacent titin filaments to form a contiguous filament along the myofibril (Figure 1a). The extensible I-band region of titin consists of the proximal and distal Ig-segments, the N2A-unique element, and the PEVK region (rich in proline [P], glutamate [E], valine [V], and lysine [K]) [22,23], and is the main contributor to passive stiffness as sarcomere length increases [24]. Long titin isoforms are less stiff and generate lower passive tensions upon sarcomere stretch. Titin also stabilizes the myofilament by limiting excessive sarcomere stretch and inhomogeneity, which is indispensable for optimal active force production [24,25]. Thus, the layout of titin in the sarcomere, together with the spring-like properties of the I-band region, make titin an ideal candidate to sense alterations in diaphragm load during mechanical ventilation.

Recently, we showed that rats deficient in the splicing factor Rbm20, leading to longer titin isoforms, are partially protected against the development of diaphragm weakness during 18 h of mechanical ventilation [19]. In addition, certain titin-binding proteins implicated in protein synthesis- and degradation pathways, such as Muscle Ankyrin Repeat proteins (MARP) and Muscle Ring finger (MuRF) 1, are upregulated in the diaphragm of mechanically ventilated critically ill patients and rodents [13,26]. The binding, and thus downstream signaling, of these proteins has been proposed to depend on titin’s mechanical state. In addition, protein binding can alter titin stiffness [26,27].

This study aimed to investigate the role of titin stiffness and its binding proteins on the development of mechanical ventilation-induced diaphragm weakness. To that end, we performed diaphragm force measurements on mice from three mouse models with genetically engineered alterations in titin stiffness after 8 h of ventilation. The findings were compared to non-ventilated mice and wild-type littermates. Based on the results in rats, we hypothesized that diaphragm weakness is attenuated in mice with a lower titin stiffness (Rbm20^ΔRRM^ mouse model) and exaggerated in mice with a higher titin stiffness (Ttn^ΔIAjxn^ and Ttn^Δ112–158^ mouse models, Figure 1a).

## 2. Results

### 2.1. Titin Mobility

Titin mobility relative to nebulin was determined to verify titin isoform size changes between the models and as a result of mechanical ventilation (Figure 1a). As expected, the diaphragm of Rbm20^ΔRRM^ mice expresses larger titin isoforms resulting in a lower mobility during gel-electrophoresis, whereas the diaphragm of Ttn^ΔIAjxn^- and Ttn^Δ112–158^ mice expresses shorter titin isoforms, resulting in a higher mobility compared to WT mice. We did not observe changes in titin mobility after 8 h of mechanical ventilation (Figure 1b).

### 2.2. The Effect of Mechanical Ventilation on Diaphragm Contractility

#### 2.2.1. Rbm20^ΔRRM^ Mice

To determine the effects of lower titin stiffness in the diaphragm, we first compared diaphragm strip contractility of non-ventilated Rbm20^ΔRRM^ mice to non-ventilated WT mice. Maximal active tension was comparable in Rbm20^ΔRRM^ and WT mice (Figure 2a). The forces at submaximal stimulations were unaltered, and so was the F50 (i.e., the frequency at which 50% of the maximal active force was reached, Figure 2b + inset, Appendix A). Furthermore, the passive tension generated and the optimal length for force production were not affected in the Rbm20^ΔRRM^ mice models (Figure 2c,d).

To assess whether lower titin stiffness affected the optimal sarcomere length, we measured the number of sarcomeres in series (i.e., from central tendon to rib) in the diaphragm strips in a subset of mice. Neither the number of sarcomeres in series in the diaphragm strips (Figure 2e), nor the tibia length (Appendix A), was significantly different in Rbm20^ΔRRM^ compared to WT mice, indicating that the optimal sarcomere length for force production was unchanged. Collectively, these results indicate that non-ventilated Rbm20^ΔRRM^ mice do not display major changes compared to WT mice.

Next, diaphragm strip contractility was assessed after 8 h of mechanical ventilation in both WT and Rbm20^ΔRRM^ mice. Maximal active tension in mechanically ventilated WT mice was 15% lower compared to non-ventilated WT mice (*p* = 0.034). In contrast, maximal active tension was comparable between mechanically ventilated and non-ventilated Rbm20^ΔRRM^ mice (Figure 2a,b). At submaximal stimulation frequencies, there was no significant effect of mechanical ventilation both in WT and Rbm20^ΔRRM^ mice (Figure 2b). The F50 was unchanged in both WT and Rbm20^ΔRRM^ mice after mechanical ventilation (Figure 2b + inset, Appendix A). Optimal length was unchanged in both WT and Rbm20^ΔRRM^ mice, but in WT mice the passive tension was higher after mechanical ventilation (*p* = 0.0003, Figure 2c,d). A higher passive tension was not observed in mechanically ventilated Rbm20^ΔRRM^ mice (Figure 2c).

Thus, in WT mice, active tension was lower and passive tension was higher after 8 h of mechanical ventilation, whereas active and passive tension in Rbm20^ΔRRM^ mice with a lower titin stiffness were preserved.

#### 2.2.2. Ttn^ΔIAjxn^ Mice

Using a similar approach, we determined whether higher titin stiffness affects contractility in diaphragm strips of Ttn^ΔIAjxn^ mice compared to WT. Maximal and submaximal active tension and F50 were comparable in non-ventilated WT and Ttn^ΔIAjxn^ mice (Figure 3a,b + inset, and Appendix A). We did not observe differences in passive tension or optimal length between Ttn^ΔIAjxn^ mice and WT mice (Figure 3c). In addition, optimal length was similar between Ttn^ΔIAjxn^ and WT mice, but diaphragm strips of Ttn^ΔIAjxn^ mice contained 12% more sarcomeres in series compared to WT (*p* = 0.0094, Figure 3d,e). Thus, non-mechanically Ttn^ΔIAjxn^ mice and WT did not differ, except that the number of sarcomeres in series was higher in diaphragm strips of Ttn^ΔIAjxn^ mice.

After 8 h of mechanically ventilation maximal active tension was reduced in WT and Ttn^ΔIAjxn^ mice by respectively 30% (*p* < 0.001) and 22% (*p* < 0.0001) compared to non-ventilated mice (Figure 3a,b). Lower active tensions were also measured at submaximal stimulation frequencies (Figure 3b). The tension-stimulation frequency relations and F50 were comparable between mechanically ventilated and non-ventilated WT mice, however in mechanically ventilated Ttn^ΔIAjxn^ mice the relation was shifted leftward, resulting in a lower F50 compared to non-mechanically ventilated Ttn^ΔIAjxn^ mice (*p* = 0.0046, Figure 3b + inset, Appendix A). Furthermore, passive tension was higher in mechanically ventilated Ttn^ΔIAjxn^ mice (*p* = 0.0095), but this was not significant in mechanically ventilated WT mice (*p* = 0.0660, Figure 3c). Mechanical ventilation did not change optimal length in Ttn^ΔIAjxn^ mice nor in WT mice (Figure 3d).

These data show that 8 h of mechanical ventilation reduces active tension to a similar extent in Ttn^ΔIAjxn^ and WT mice. Thus, higher titin stiffness, induced by deleting the IA junction, does not exaggerate mechanical ventilation-induced diaphragm weakness.

#### 2.2.3. Ttn^Δ112–158^ Mice

To study the effects of a severely increased titin stiffness, we measured diaphragm strip contractility in Ttn^Δ112–158^ mice and compared the results to WT mice. Maximal active tension was comparable between Ttn^Δ112–158^ mice and WT mice (Figure 4a,b). However, the tension-frequency relation shifted rightward in Ttn^Δ112–158^ mice, increasing F50 (*p* = 0.0113, Figure 4b + inset, and Appendix A). Similar as in Ttn^ΔIAjxn^ mice, we did not observe an increased passive tension at optimal length in Ttn^Δ112–158^ mice, compared to WT mice (Figure 4c). The optimal length of Ttn^Δ112–158^ mice diaphragm strips was 11% longer compared to WT mice (*p* = 0.0094, Figure 4d) and strips contained 34% more sarcomeres in series (*p* < 0.0001, Figure 4e).

In line with the findings in the Ttn^ΔIAjxn^ mice, maximal active tension was reduced in both mechanically ventilated WT and Ttn^Δ112–158^ mice by respectively 29% (*p* = 0.0001) and 21% (*p* = 0.0004) compared to non-ventilated mice (Figure 4a,b). A lower active tension was also present at submaximal stimulation frequencies (Figure 4b). No significant changes were observed in the F50 after mechanical ventilation in either WT or Ttn^Δ112–158^ mice (Figure 4b + inset, Appendix A). Like in Ttn^ΔIAjxn^ mice, passive tension was increased after mechanical ventilation in Ttn^Δ112–158^ mice (*p* = 0.0001), and in ventilated WT mice (*p* = 0.0069, Figure 4c). Mechanical ventilation did not affect the optimal length in WT and Ttn^Δ112–158^ mice (Figure 4d).

These data show that active tension is lower in both WT and Ttn^Δ112–158^ after 8 h of mechanical ventilation, and that severely increased titin stiffness does not exaggerate the contractile dysfunction.

### 2.3. Titin Binding Proteins

#### 2.3.1. Muscle Ankyrin Repeat Proteins (MARP’s)

Next, we studied whether differential expression of titin binding proteins in the mouse models can explain the preserved diaphragm contractility in ventilated Rbm20^ΔRRM^mice. We first focused on MARP proteins as MARP1 is highly upregulated in mechanically ventilated critically ill patients and rats, and is known to increase passive tension upon stretch as it tethers the N2A region in titin’s spring region to the thin filament [26]. Additionally, MARP1 is differentially expression upon passive stretch in Ttn^ΔIAjxn^ mice compared to Rbm20^ΔRRM^ mice and may be involved in the hypertrophic response [26].

MARP1 and -2 expression were assessed by western blot, but MARP1 could not be detected in all samples. MARP1 levels were higher in non-ventilated Ttn^Δ112–158^ mice compared to WT (*p* = 0.009) but did not significantly change after mechanical ventilation (Figure 5a). MARP2 expression was higher in non-ventilated Rbm20^ΔRRM^ mice compared to non-ventilated WT (*p* < 0.0001) and reduced after mechanically ventilated (*p* = 0.0036, Figure 5b). No other changes in MARP1 and -2 expressions were observed. Thus, MARP 1 and 2 expression were altered in non ventilated Ttn^Δ112–158^ and Rbm20^ΔRRM^ mice compared to their WT littermates. These changes may contribute to diaphragm adaptation during mechanical ventilation, and to the preservion of active tension in mechanically ventilated Rbm20^ΔRRM^ mice.

#### 2.3.2. Muscle RING-Finger Protein-1 (MuRF1)

MuRF1 is another titin binding protein and a critical component of the ubiquitin-proteasome pathway. This pathway is activated in mechanically ventilated critically ill patients and ventilated animals and contributes to diaphragm atrophy and contractile weakness [13]. Moreover, mice lacking MuRF1 expression did not develop diaphragm weakness [13]. MuRF1 expression in non-ventilated mice was comparable to WT in all three models. After mechanical ventilation MuRF1 expression was higher in both WT (*p* = 0.0407) and Rbm20^ΔRRM^ mice (*p* = 0.0239) and in Ttn^ΔIAjxn^ mice (*p* = 0.0465). In the Ttn^Δ112–158^ model, MuRF1 expression was significantly higher in WT mice only (*p* = 0.0044) (Figure 5c). These findings show that the ubiquitin-proteasome pathway is activated in titin models with a decreased and increased titin stiffness, and therefore most likely does not play a role in preserving active and passive tension in Rbm20^ΔRRM^ mice.

#### 2.3.3. Calpain 3

Finally, we determined protein abundance of both full-length and autolytic calpain3 (active calpain-3) in the diaphragm of the mice. The role of this protein in diaphragm adaptation following mechanical ventilation is unclear, but it plays well established roles in apoptosis pathways in muscle. The ratio of autolytic calpain3 to full length calpain3 was not different from WT mice in all three models. Mechanical ventilation did not affect the ratio of autolytic calpain3 to full length calpain3 in any of the models (Figure 6a). In addition, expression of full-length and autolytic calpain3 normalized to total protein did not significantly differ from WT mice in any of the models, neither in ventilated nor in non-ventilated mice (Figure 6b,c). Thus, calpain3 expression/activation is not affected by titin stiffness and mechanical ventilation.

## 3. Discussion

This study aimed to investigate the role of titin stiffness and its binding proteins in the development of mechanically ventilation-induced diaphragm weakness. By ventilating mice with altered titin stiffness, we showed that diaphragm active tension was preserved during ventilation in Rbm20^ΔRRM^ mice with low titin stiffness. Ttn^ΔIAjxn^ and Ttn^Δ112–158^ mice, expressing stiff titin isoforms, showed a reduced active tension after mechanical ventilation, comparable to WT mice. Thus, stiff titin isoforms did not exaggerate diaphragm weakness after mechanical ventilation. Analyses of titin-binding proteins suggest a role for MARP 1 and 2 in the adaptation of the diaphragm to mechanical ventilation, and to the preservion of active tension in Rbm20^ΔRRM^ mice.

### 3.1. Effects of Titin Stiffness on Active Tension in Non-Ventilated Mice

Titin’s elastic properties are essential for active and passive force production. It maintains the A-band centered in the sarcomere and secures myofilament integrity during mechanical loading and stretch by connecting the thin and thick filament [25]. Titin-based passive tension increases as sarcomeres lengthen, bringing actin and myosin in closer proximity and thus reducing the myofilament lattice spacing. This facilitates actomyosin interaction [24,25]. Since titin-based passive tension is directly related to lattice spacing [28,29], we hypothesized that alterations in titin stiffness would affect force production. Here, we show that modifying titin stiffness does not affect the passive and active tension measured at optimal length in diaphragm strips of non-ventilated mice (Figure 2, Figure 3 and Figure 4a,b).

Possibly, the sarcomere length at optimal diaphragm strip length in each of the three models was such that passive tension within the sarcomeres was not significantly different, and therefore lattice spacing and actomyosin interaction were not affected across the models.

In the Rbm20^ΔRRM^ and Ttn^ΔIAjxn^ mice, sarcomere length at optimal strip length was 2.9 ± 0.2 μm and 3.0 ± 0.1 μm respectively. At these sarcomere lengths no major differences in passive tension have been reported [30]. Similarly, in the Ttn^Δ112–158^ mice, the sarcomere length at optimal strip length was ~2.3 ± 0.2 µm, a length at which passive tension is expected to be comparable to that generated at Lo and its associated sarcomere length (2.9 ± 0.2 μm) in the WT mice [31]. Indeed, this suggests that the number of sarcomeres in series was adapted to maintain sarcomere length in a range at which passive tension was minimal, and at which no major differences in passive tension are present between the three models. Moreover, we anticipate that also in vivo the sarcomeres in the diaphragm of the three models do not experience major differences in passive tension. This anticipation is based on recent work that showed that passive tension in diaphragm fibers of the Rbm20^ΔRRM^ and Ttn^ΔIAjxn^ mice only differed from WT at sarcomere lengths > 3.5 µm [30], while the in vivo working range in the diaphragm during tidal breathing in WT, Rbm20^ΔRRM^, and Ttn^ΔIAjxn^ mice is approximately 2.2–2.9 µm [30] and in Ttn^Δ112–158^ mice 1.9–2.4 µm due to a compensatory increase in the number of sarcomeres in series [31]. Thus, both when measured ex vivo at optimal length and in vivo during tidal breathing, there is likely no difference in diaphragm passive tension between the models.

Interestingly, MARP1 was upregulated in non-ventilated Ttn^Δ112–158^ mice, exhibiting the stiffest titin isoform, whereas MARP2 was upregulated in non-ventilated Rbm20^ΔRRM^ mice, expressing the most compliant isoform. MARP1 tethers the N2A region in titin’s spring region to the thin filament. Filament [26]. Likely, MARP1 upregulation in Ttn^Δ112–158^ mice contributes to the stiffer titin molecules, but which is compensated by the shorter sarcomere working range and the increase in sarcomeres in series. The function of MARP2 is not clear, but it shares over 50% homology with MARP1. If MARP2 has a similar tethering function as MARP1, upregulation in Rbm20^ΔRRM^ non-ventilated mice may cause increased tethering of titin to actin. This could possibly serve to ‘restress’ titin to its physiological, shorter lengths without increasing sarcomere length and working range to restore passive tension.

An additional explanation for the unchanged force production in the different titin models might relate to titin-based passive tension varying among muscles and depending on the size of the titin isoforms expressed. As the diaphragm expresses a relatively large titin isoform (3700 kDa, [32]), increases in titin isoform size due to Rbm20 deficiency in rats or deletion of the RNA recognition motif of Rbm20 in mice may have a relatively minor impact on force production compared to the same genetic alteration in cardiomyocytes or other skeletal muscle containing smaller titin isoforms [19,33,34,35]. This is in line with previous work in Ttn^ΔIAjxn^ mice, which showed diastolic dysfunction in the absence of skeletal muscle dysfunction [36]

### 3.2. Effect of Titin Stiffness on Active and Passive Tension during Mechanical Ventilation

Although changes in titin-based stiffness did not affect diaphragm contractility in non-ventilated mice, we were motivated to investigate whether maximal active tension was preserved in Rbm20^ΔRRM^ mice after 8 h of mechanical ventilation, as was observed previously in 18 h- ventilated Rbm20 deficient rats [19]. In line with the findings in rats, active tension was preserved in ventilated Rbm20^ΔRRM^ mice (Figure 2a). As reduced titin stiffness preserves diaphragm function, we anticipated that mice with stiffer titin isoforms may have exaggerated ventilation-induced diaphragm weakness. However, mechanically ventilated Ttn^ΔIAjxn^ and Ttn^Δ112–158^ mice developed diaphragm weakness to a similar extent as ventilated WT mice. This suggests that titin stiffness per se does not modulate the effect of mechanical ventilation on diaphragm contractility, but that titin bindings proteins may play a role. The binding of these proteins, e.g., MuRF-1 and Calpain3, depends on the mechanical state of titin and may contribute to contractile dysfunction in ventilated mice [13,27]. However, in our study, MuRF-1, Calpain3 as well as MARP1 were not differentially regulated in Rbm20 ^ΔRRM^ mice versus Ttn^ΔIAjxn^—and Ttn^Δ112–158^ mice, and therefore these proteins are unlikely to be involved in the preservation of diaphragm contractility in ventilated Rbm20^ΔRRM^ mice.

MARP2 expression, however, was higher in non-ventilated Rbm20^ΔRRM^ mice compared to WT mice, and decreased during mechanical ventilation, while it remained unchanged in the other models. Based on our data, we can only speculate as to whether there is a causative relationship between MARP2 expression and the preservation of active tension during mechanical ventilation in Rbm20^ΔRRM^ mice.

If MARP2 indeed tethers the N2A region to actin, the reduction in MARP2 expression during mechanical ventilation could help to adapt the diaphragm fibers by lowering titin compliance to preserve contractility. Such a mechanism would be only possible to a limited extent in the Ttn^ΔIAjxn^ and Ttn^Δ112–158^ mice where the strain on titin is already at or beyond physiological levels. However, we cannot fully rule out that the protective effects of the Rbm20^ΔRRM^ may involve changes in the splicing of other targets, as it was shown that Rbm20 regulates isoform expression of several genes, involved in sarcomere assembly, ion transport and diastolic function in the heart [35]. Whether or not proteins of the MARP family can help regulate titin stiffness to preserve diaphragm contractility upon mechanical ventilation should be the subject of further investigations.

### 3.3. Mechanical Ventilation and Clinical Implications

Based on previous studies from our group [13], 8 h of ventilation is sufficient to induce diaphragm weakness in mice, without enduring the hemodynamic instability and mortality that come with a longer ventilation duration [37,38]. In general, small laboratory animals have a higher metabolism and a faster protein turnover, and we speculate that for this reason the effects of mechanical ventilation develop more rapidly compared to humans. The mice in our study received controlled mechanical ventilation and diaphragm activity was completely suppressed by the ventilator, whereas mechanically ventilated patients often display some low-level diaphragm activity during prolonged ventilation. Thus, the mouse model resembles the complete disuse phase of mechanical ventilation, as occurs during deep sedation.

Although the exact mechanism remains unclear, our data show that reduced titin stiffness preserves diaphragm contractility during mechanical ventilation. Clinically induced modifications in titin compliance in the context of mechanical ventilation would only be feasible if (1) they can be administered specifically to the diaphragm, (2) have no (cardiac) side effects, and (3) act reasonably quick. Clearly, genetic approaches do not meet these criteria yet and appear far from clinical application. In addition to genetic approaches, posttranslational modifications (PTMs) such as oxidation/reduction, (de)acetylation, and (de)phosphorylation are known to alter titin compliance [39] and allow for quick and reversible adjustments. However, the enzymes responsible for the aforementioned PTMs are not specific to titin, and clinical trials that targeted disrupted phosphorylation in heart failure with pathologically increased myocardial passive stiffness have not been successful so far [39].

An interesting recent development is the use of antisense oligonucleotides (ASO’s). ASO’s interfere with messenger RNA (mRNA) expression of their target through a variety of mechanisms, including inhibition of mRNA translation [40]. Radke et al. used ASO’s to reduce the expression of Rbm20 in a mouse model of heart failure with preserved ejection fraction (HFpEF) [41]. Weekly application of cardiac Rbm20 specific ASO’s for 8 weeks caused a 50% reduction in RBM20 expression and a switch in titin isoform expression to more compliant isoforms, both in WT mice and N2B KO mice with increased titin-based stiffness. ASO-treated mice showed improved cardiac function, without major side effects. The translational potential of this treatment is supported by the finding that three weeks of ASO treatment in human engineered heart tissue led to a similar 50% reduction in RBM20 expression, along with improved relaxation- and contraction kinetics [41]. To make this approach feasible in the setting of mechanical ventilation, concerns regarding the required duration and specificity of this technique will need to be addressed. While 8 weeks of ASO treatment in mice, and three weeks in human engineered heart tissue, was sufficient to downregulate RBM20 expression by 50%, mechanical ventilation-induced diaphragm weakness is often already present within 24 h [1]). Thus, future studies should address whether ASO treatment can reverse the development of diaphragm weakness, like the findings in the heart failure models. In addition, the ASO’s were screened and selected for their efficiency in the heart, with minimal hepatotoxicity, but also reduced RBM20 expression in skeletal muscle. Directing ASO’s specifically to the diaphragm, but not to other skeletal muscles, will likely be challenging.

In conclusion, reduced titin stiffness protects against mechanical ventilation-induced diaphragm weakness. The mechanisms underlying this protective effect may involve titin-based mechanosensing through members of the MARP protein family. Future research should focus on exploring these mechanisms to develop new therapeutic targets to treat diaphragm weakness in ventilated ICU patients.

## 4. Materials and Methods

### 4.1. Animal Models

Three genetically engineered mouse models expressing longer or shorter titin isoforms were used. The first model was created through deletion of the RNA recognition motif (exon 6&7) of the splicing factor Rbm20 (Rbm20^ΔRRM^) [33]. This results in longer, less stiff titin molecules that generates lower passive tensions at a given sarcomere length [30]. Second, deletion of the IA junction of titin’s spring region by removal of exon 251–269 (Ttn^ΔIAjxn^, [36]) results in a smaller and mildly stiffer titin isoform. Finally, an even stiffer titin isoform was created by removing exon 112–158, thereby deleting a large part (~167.1 kDa in mouse diaphragm) of the PEVK segment (Ttn^Δ112–158^), the most flexible spring region in titin [31] (Figure 1a). All mice were on a C57BL/6j background. Maximal active tensions and passive tensions were similar among non-ventilated wild-type mice of all three models (Appendix A). Homozygous mice and their wild-type (WT) littermates were used for experiments at 4 months of age.

All experiments were performed in accordance with the University of Arizona Institutional Animal Care and Use Committee (protocol number 13-488) and following the US National Institutes of Health Using Animals in Intramural Research guidelines for animal use.

### 4.2. Mechanical Ventilation

Mice were anesthetized with a mixture of ketamine, dexmedetomidine, and atropine (127.5/0.1/0.5 mg/kg) in lactated ringer’s solution I.P., tracheostomized, and connected to the ventilator (PhysioSuite, Kent Scientific, Torrington, CT, USA). Mice were mechanically ventilated for 8 h using O_2_ enriched air (40%) using the following settings: tidal volume 7 mL/kg body weight, respiratory rate 150/min, I:E ratio, 1:2, and 3 cmH_2_O positive end-expiratory pressure.

Anesthesia was maintained throughout the experiment with ketamine, dexmedetomidine, and atropine (36/0.02/0.1 mg/kg/h) in lactated ringer’s solution I.P. Anesthesia depth was monitored using: heart rate, etCO_2_, toe pinch reflex, and a 30-s end-expiratory respiratory hold was applied to verify that the breathing reflex was adequately suppressed. Where necessary, anesthesia depth was adjusted. In addition to the anesthetics, alternately 100 uL lactated ringer’s solution and sodium bicarbonate solution (1 mL 8.4% sodium bicarbonate in 5 mL lactated ringer’s solution) were administered hourly to maintain an adequate fluid balance monitored by urine production. Body temperature was kept at 37 °C using an automated feedback system. Non-mechanically ventilated mice were anesthetized using isoflurane. All mice were sacrificed by cervical dislocation before excision of the diaphragm.

### 4.3. Intact Mechanics

The contractile function of intact diaphragm muscle strips was measured as previously described [42]. Immediately after sacrificing the animals, the diaphragm was removed. A diaphragm strip, including tendon and ribs, was excised from the central costal region of the right hemidiaphragm at the insertion of the phrenic artery. Using silk sutures, the strip was tightly sutured and mounted vertically in a tissue bath between a dual-mode lever arm and a fixed hook (1200A Intact Muscle Test System, Aurora Scientific Inc., Aurora, ON, Canada). All experiments were performed in oxygenated (95% O_2_, 5% CO_2_) mammalian Ringer’s solution (pH 7.40, 30 °C).

First, maximal active twitch force was measured to determine optimal length. Subsequently, the maximal tetanic force was determined using a 0.3-s tetanus at 150 Hz. The passive force was measured at the baseline of the maximal active force trace at optimal length. Three minutes after determination of maximal tetanic force the muscle strips were stimulated at incremental stimulation frequencies (1, 5, 10, 20, 30, 40, 60, 80, 100, 150 Hz) for 600–1000 ms, with a 30–120 s interval. After completion of all measurements, the length and weight of the diaphragm strips were determined and used to calculate the cross-sectional area, taking into account a specific density of 1.056 g/mL. Data analysis for active force was performed on the highest forces measures throughout the protocol. Force was normalized to muscle cross-sectional area in mN/mm^2^ (i.e., active or passive tension at optimal length).

Individual force-frequency relations were normalized to maximal active tension and a non-linear curve was fitted through the data. The top was constrained at 100%, while the baseline was set to be 0%. The absolute IC50 was calculated to determine the frequency at which 50% of the maximal force was generated (F50, a parameter which depends on fiber type composition and calcium handling). In some diaphragm strips, force-frequency relations displayed minor rundown (<10% of the maximal active force). These data were included in the analysis, and no difference in rundown between the groups was observed.

Note that we could not measure contractility and the number of sarcomeres in series in the same diaphragm strip. Therefore, sarcomeres in series were measured in a subset of mice using diaphragm strips excised based on the same anatomical landmarks (insertion of the phrenic artery from the from the central costal region of the right hemidiaphragm) as was used for contractility measurements.

### 4.4. Protein Analyses

The ventral part of the diaphragms right costal was flash-frozen in liquid nitrogen and stored at −80 °C until further use. Tissue was weighed and ground fine using Dounce homogenizers cooled in liquid nitrogen. Subsequently, samples were acclimated to −20 °C for 40–60 min and suspended in homogenization buffer (1:1 *v*/*v* mixture of [8 M Urea, 2 M thiourea, 0.05 M Tris-HCl, 0.075 M dithiothreitol, 1.5% SDS, 0.03% bromophenol blue, pH 6.8]:[50% glycerol containing 0.04 mM E-64, 0.16 mM leupeptin, and 0.2 mM PMSF]. Samples were mixed and incubated at 60 °C for 14 min. Homogenates were centrifuged at 13.500 rpm for five minutes and the supernatant was collected.

4–20% bis-acrylamide gels were run with equal loading for all samples and stained using Neuhoff’s Coomassie brilliant blue staining protocol. Gels were scanned using a commercial scanner (Epson 800, Epson Corporation, Long Beach, CA, USA). Myosin Heavy Chain (MyHC) levels were determined using Image Studio Lite v5.2.5 (Li-cor Biosciences, Lincoln, NE, USA). Samples were diluted to equalize MyHC content and run on 10% SDS-PAGE gels at 100 V for two hours and semi-dry transferred onto Immobilon-P PVDF 0.45 μm membranes (Millipore, Billerica, MA, USA) using a Trans-Blot Semi-Dry transfer system (Bio-Rad, Hercules, CA, USA) at 123 mA/gel for 50 min. Membranes were stained using Ponceau S (Bio-Rad, Hercules, CA, USA) and dried overnight. After scanning (Epson 800, Epson Corporation, Long Beach, CA, USA), membranes were re-hydrated in 100% methanol and blocked for 1 h in 1:1 *v*/*v* PBS:Odyssey PBS blocking buffer. Membranes were incubated with primary antibody at 4 °C overnight (αMuRF1, 1:1000; αCAPN3, 1:500; αAnkrd1 (MARP1), 1:500; αAnkrd2 (MARP2), 1:4000, all from Myomedix, Neckargemünd, Germany). A secondary antibody was applied for 1 h at RT (goat anti-rabbit CF680/790, [#20067] or goat anti-chicken CF660 [#20126], both 1:20,000, from Biotium, Fremont, CA, USA]. Blots were imaged using the Odyssey Infrared Imaging System (Li-Cor Biosciences, USA) and signal intensity was determined using Image Studio Lite. Protein expression levels were normalized against total protein in the corresponding sample, as determined from Ponceau S using Image J 1.53 k (National Institutes of Health, USA). A pooled sample was run on all gels to correct for the use of multiple blots.

### 4.5. Titin Mobility

SDS- agarose gel electrophoresis (SDS-AGE) was used to determine changes in titin mobility relative to nebulin, as described before [30]. In short, samples were run on a 1.2% agarose gel for 4 h at 15 mA per gel and stained using Neuhoff’s Coomassie brilliant blue staining protocol.

### 4.6. Statistics

Graphpad Prism v9.3.1 was used for statistical analysis. Data were checked for normality using the Shapiro-Wilk test. In the case of non-normality, data were log-transformed or square root-transformed. The three models were analyzed as separate experiments, each with their own wild-type littermates as control. For each model, the following three comparisons were made: wild-type vs model, wild-type vs mechanically ventilated wild-type, and model vs mechanically ventilated model, and corrected for multiple comparisons using the Šídák correction or Dunn’s test when transformations did not resolve non-normality. Data are presented as mean ± SEM.

## Figures and Tables

**Figure 1 ijms-23-15689-f001:**
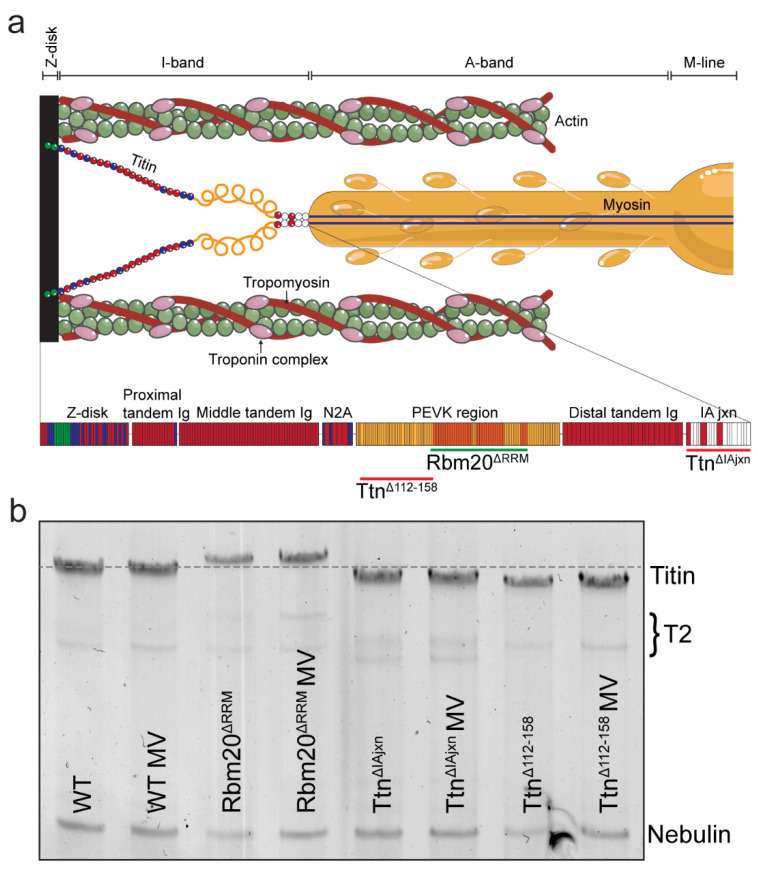
**Titin migration for the different experimental models.** (**a**) Layout of titin in the sarcomere. The N-terminal end of titin is anchored to the Z-disc, while the C-terminal end is anchored to myosin at the M-band. Below is a closer representation of the spring region of titin, between the Z-disc and IA-junction. The extra PEVK segments in the Rbm20^ΔRRM^ model are the orange domains, underlined in green. The deletions in the Ttn^ΔIAjxn^ and Ttn^Δ112–158^ mice are underlined in red. (**b**) Titin migration relative to nebulin, as determined by SDS-AGE is shown for Rbm20^ΔRRM^, Ttn^ΔIAjxn^ and Ttn^Δ112–158^ mouse diaphragm respectively, for both non-ventilated and ventilated mice. A dotted gray line on the height of WT is indicated for reference. T2 indicates titin degradation products. For clearer representation of titin, contrast of the original scan was enhanced. The enhanced picture is shown here.

**Figure 2 ijms-23-15689-f002:**
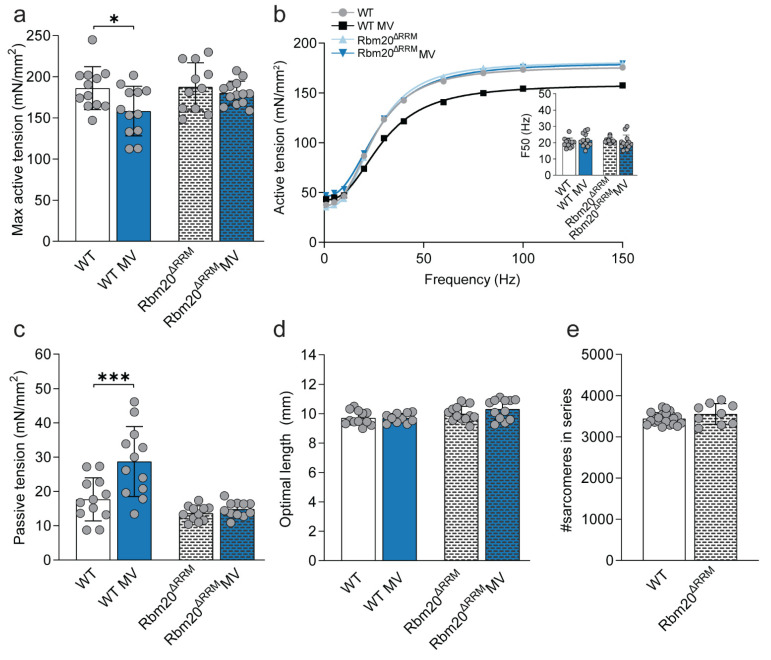
**Effects of mechanical ventilation on muscle function in the Rbm20^ΔRRM^ mouse diaphragm.** (**a**) Maximal active tension of non-ventilated and ventilated WT and the Rbm20^ΔRRM^ mice. (**b**) Fitted curves for tension-stimulation frequency relations. F50 (inset) was determined as the absolute EC50 of the curve, and represents the frequency required to develop 50% of maximal force. (**c**) Passive tension measured at optimal length in non-ventilated and ventilated WT and Rbm20^ΔRRM^ mice. (**d**) Optimal diaphragm strip length, i.e., the length representing maximal cross bridge binding required for maximal force production. (**e**) Number of sarcomeres in series. Bars represent mean ± SD. * *p* < 0.05, *** *p* < 0.001. WT = wild type, MV = mechanically ventilated. Each dot represents a single animal and animals are only presented in each graph once.

**Figure 3 ijms-23-15689-f003:**
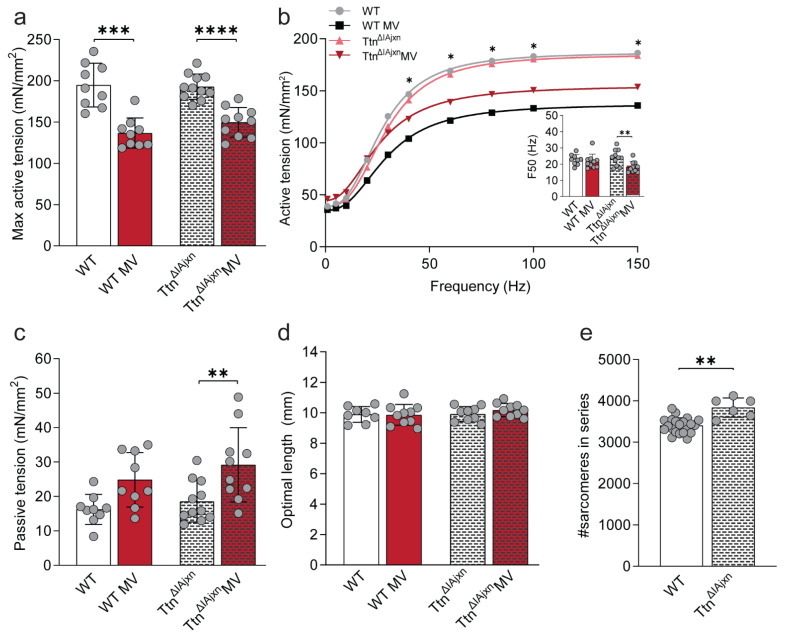
**Effects of mechanical ventilation on muscle function in the Ttn^ΔIAjxn^ mouse diaphragm.** (**a**) Maximal active tension of non-ventilated and ventilated WT and the Ttn^ΔIAjxn^ mice. (**b**) Fitted curves for tension-stimulation frequency relations. F50 (inset) was determined as the absolute EC50 of the curve, and represents the frequency required to develop 50% of maximal force. Asterisks indicate a significant reduced force in mechanical ventilated animals compared to their respective non-mechanically ventilated controls. (**c**) Passive tension measured at optimal length in non-ventilated and ventilated WT and Ttn^ΔIAjxn^ mice. (**d**) Optimal diaphragm strip length, i.e., the length representing maximal cross bridge binding required for maximal force production. (**e**) Number of sarcomeres in series. Bars represent mean ± SD. * *p* < 0.05, ** *p* < 0.01, *** *p* < 0.001, **** *p* < 0.0001. WT = wild type, MV = mechanically ventilated. Each dot represents a single animal and animals are only presented in each graph once.

**Figure 4 ijms-23-15689-f004:**
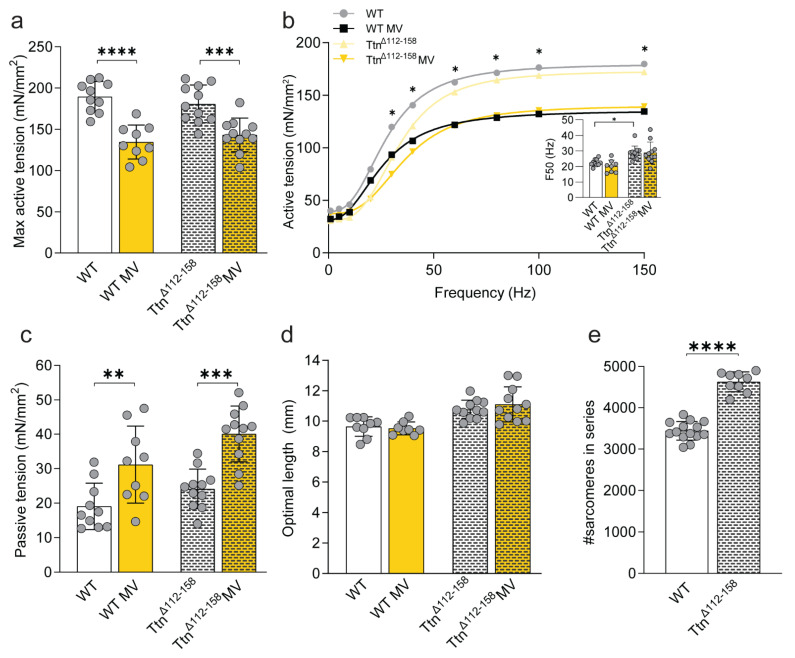
**Effects of mechanical ventilation on muscle function in the Ttn^Δ112–158^ mouse diaphragm.** (**a**) Maximal active tension of non-ventilated and ventilated WT and the Ttn^Δ112–158^ mice. (**b**) Fitted curves for tension-stimulation frequency relations. F50 (inset) was determined as the absolute EC50 of the curve, and represents the frequency required to develop 50% of maximal force. Asterisks indicate a significant reduced force in mechanical ventilated animals compared to their respective non-mechanically ventilated controls. (**c**) Passive tension measured at optimal length in non-ventilated and ventilated WT and Ttn^Δ112–158^ mice. (**d**) Optimal diaphragm strip length, i.e., the length representing maximal cross bridge binding required for maximal force production. (**e**) Number of sarcomeres in series. Bars represent mean ± SD. * *p* < 0.05, ** *p* < 0.01, *** *p* < 0.001, **** *p* < 0.0001. WT = wild type, MV = mechanically ventilated. Each dot represents a single animal and animals are only presented in each graph once.

**Figure 5 ijms-23-15689-f005:**
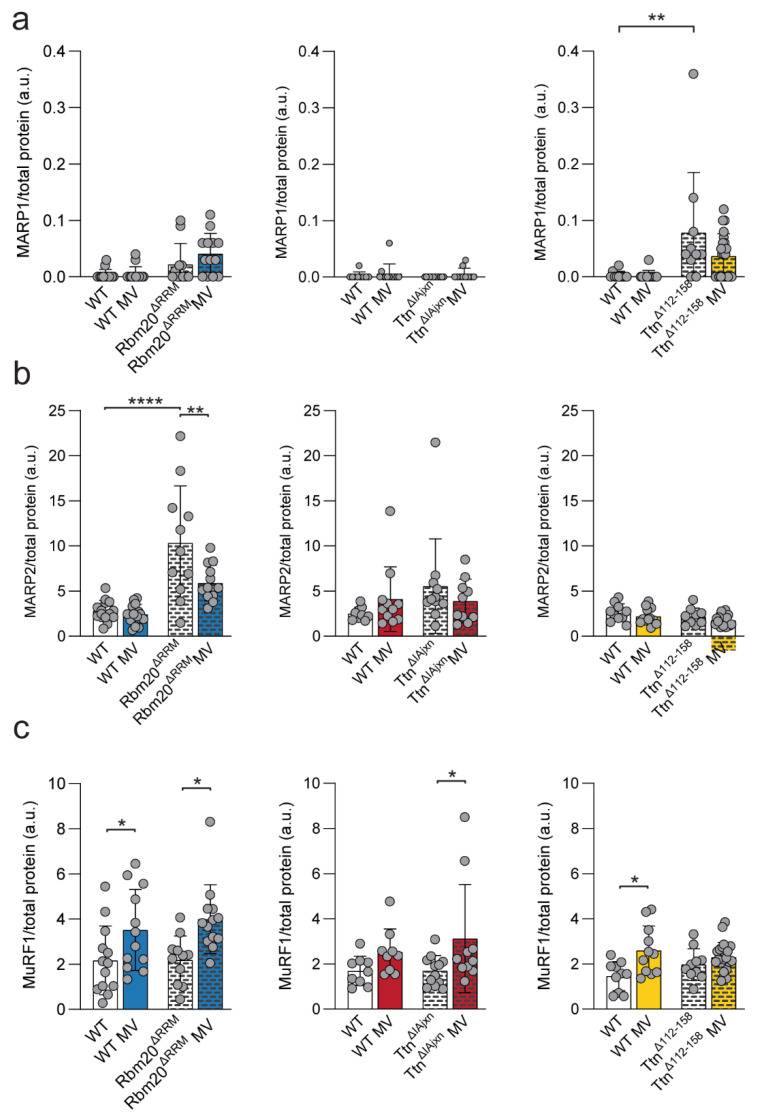
**Protein expression levels of MARP1, 2 and MuRF1.** (**a**–**c**) MARP1, MARP2, MuRF1 protein expression relative to total protein in Rbm20^ΔRRM^, Ttn^ΔIAjxn^ and Ttn^Δ112–158^ mice respectively. Bars represent mean ± SD. * *p* < 0.05, ** *p* < 0.01, **** *p* < 0.0001. WT = wild type, MV = mechanically ventilated. Each dot represents a single animal and animals are only presented in each graph once.

**Figure 6 ijms-23-15689-f006:**
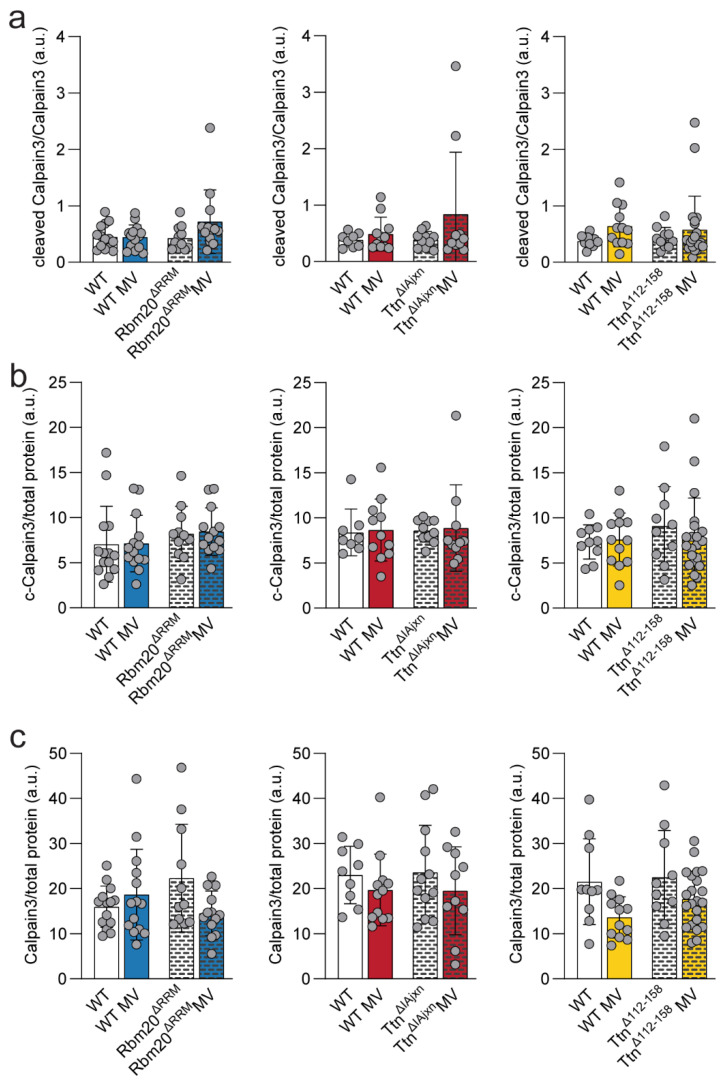
**Protein expression levels of autolytic calpain3/calpain3 ratio, cleaved calpain3, and calpain3.** (**a**) Cleaved calpain/calpain 3 ratios, (**b**) cleaved calpain3, and (**c**) calpain3 protein expression relative to total protein in Rbm20^ΔRRM^, Ttn^ΔIAjxn^ and Ttn^Δ112–158^ respectively. Bars represent mean ± SD. WT = wild type, MV = mechanically ventilated. Each dot represents a single animal and animals are only presented in each graph once.

## Data Availability

Any data or material that support the findings of this study can be made available by the corresponding author upon request.

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
