# Peer review of "Rbm20ΔRRM Mice, Expressing a Titin Isoform with Lower Stiffness, Are Protected from Mechanical Ventilation-Induced Diaphragm Weakness"

_ijms, 2022, doi:10.3390/ijms232415689_

Round 1

Reviewer 1 Report

This study by van den Berg et al. compares three different titin mutation (or splice variation) mouse models.  One is the RBM20 model, which has longer or more compliant titin.  The other two models are shorter or stiffer titin molecules in the sarcomere.  The focus of the study is diaphragm function following mechanical ventilation, and how these different titin variants either benefit or reduce deteriorating diaphragm function.  This is a very important clinical problem, in that mechanical ventilation compromises diaphragm function and multiple ventilation bouts becomes very dangerous and deadly because the contractility of the diaphragm does not recover well upon mechanical ventilation.  The authors suggest that the RBM20 model may be a direction showing that more compliant diaphragm tissue or more compliant titin may afford better diaphragm function following mechanical ventilation.  The stiffer titin, and wild type animals show detrimental diaphragm contraction following 8 hours of mechanical ventilation.

The mechanics measurements seem pretty clear and direct.  There is some confusion or lacking justification for the biochemistry for the 3 proteins of interest that were investigated.  There is not a problem with the data, there is just some questions about why it was these 3 proteins vs. other proteins of interest.  I realize they are all titin binding proteins, or involved in the ubiquitin proteosome pathway that is activated in mechanically ventilated critically ill patients (i.e. MuRF1).  I’m even a bit more confused about the Calpain 3 focus, although it is a portion of the apoptosis pathway, which could contribute to the muscle wasting or dysfunction—but it doesn’t appear to be involved.  Aneckdotally, I think all the data presented is good, but may need a bit more clarification on justification of the proteins of interest.

I applaud the authors work with the terrific supplemental information and clear presentation of the whole gels that were used.  The graphical overview is also fantastic, and really ties the important aspects of the study together all in one clear figure.

Specific Concerns or Comments:

Line 17—“mechanosensing”.  Are the authors simply saying that titin is an important protein that affects sarcomere mechanics, or is there some activation or sensation and signaling component of mechanosensation.  This may be trivial, but could be clarified as well.

Line 27--The focus on MARP2 helping mechanical ventilation in the RBM20 because it is upregulated or more concentrated vs. other protein expression in this titin mouse model may be true, but may be a little misleading for what is also shown for the MARP1 pathway being detrimental in the titin deletion 112-158 model.  Aren’t both important or worth commenting upon (or reducing the claim on MARP2 without juxtaposition of MARP1)?

Can the authors comment or strengthen the idea about mechanical ventilation for 8 hours?  This comes up first in the graphical abstract, but also in the discussion (Sec. 3.2, and 3.3) and Methods.  How similar is the protocol utilized in this study translatable or comparable to other rat and mouse ventilation conditions.  And how do these rodent protocols compare with or emulate human ventilation protocols, are they focused on acute ventilation only or does it have more broad applications?

Line 85—MV is used, but I don’t see previously, nor here, where MV=mechanical ventilation is introduced.

Bottom line of Fig. 1 legend: “For clearer representation of titin, contrast of the picture was enhanced equally for the whole gel shown here.”  There can be a little more clarity here that the ‘enhanced’ means enhance compared to the image shown in the supplemental material.  AS currently written, not so clear.

In all the Mechanics figures, there is panel e representing the number of sarcomeres in series.  What is this supposed to represent?  Does it contribute to the mechanics, or the dysfunctional contraction, or is it simply a metric of morphological differences among the three models?  Along the same vein, why are there so few data points in Fig. 3e, for the delta IA junction data?  How does the findings here in the diaphragm compare to the sarcomere length in series from the muscles tested in the PNAS paper where the IA junction mouse was first introduced?

Figure 5—It is a little unclear how the “protein of interest” to “total protein” ratios are calculated or measured.  In addition, the differences between MARP1 and MARP2 may afford some flexibility to be more greatly discussed between the “stiffest” and the “most compliant” models.

Can the authors clarify when or how the passive vs. maximal active tension was measured?  Why is the maximal tension different from the traces that were fit to the curves?  What is the significance of a higher or lower F50, related to normal muscle function and disease?

Are there any differences of setting optimal length at maximal twitch force vs. setting optimal length a maximal tetanic (minus passive) force?  How are the tissues attached to the equipment, related to concerns on compliance that may shift optimal length differently between a twitch vs. a tetanic contraction? Would this affect any of the major conclusions or findings?

Lines 238-242 suggest that there may be some differences or divergence between changes in lattice spacing and changes in passive and active force.  This could be an insightful component, which may be unique to the titin models—or more general.  Can the authors comment on this divergence or think this is an important avenue to pursue further?

Discussion lines 243-260.  This is an insightful and strong paragraph, but a little terse and confusing for those not up to speed on the different effects of titin on different components of sarcomere length vs. mechanical contraction.  Reworking would be helpful, with more step by step detail in understanding.

Discussion lines 261-262.  This first sentence is a little confusion about the details it is referring to.  From my looking at the data, there really are not many differences in passive forces (20-30 mN/mm2 out of 200 or 250 maximally activated=~5% differences), aside from the MV deletion112-158 data?  Please clarify on what could represent physiologically important differences in passive stiffness between models.

Lines 296-301.  I’m not clear on the “restress”, would de-stress be a better description?  Then to follow up with why the two stiffer mouse models would not be possible to share a similar mechanism because it is already beyond physiological strain is not clear.  Please clarify or revised the descriptions of this few lines.

Reviewer 2 Report

This manuscript is a concise and elegant study on how three genetic modifications affecting titin modulate the biomechanical properties of the diaphragm upon mechanical ventilation. Authors report that Rbm20ΔRRM mice do not develop increased passive stiffness and retain contractility upon mechanical ventilation and correlate this phenomenon to increased expression of MARP2.

Major concern

  1. My biggest concern about this study is significant internal variation, casting doubt about the robustness of article claims. Depending on the experiment, conceptually identical mechanical ventilation of WT mice either caused or did not cause an increase in passive tension. Authors need to provide additional evidence that observed differences are not merely due to batch effects. Authors could address this issue by analyzing datasets together and introducing batch as a random factor. Calculating, reporting, and discussing effect sizes would allow formal evaluate the impact of different titin genotypes on physiological parameters upon mechanical ventilation. 

Minor comments

  1. Authors should provide WB data.
  2. Clearly define the experimental unit in each graph.
  3. Did the authors adjust for possible pseudoreplication if several samples/measurements were derived from a single animal/specimen?
  4. Authors should discuss the possibility that the protective effects of the Rbm20ΔRRM genotype could be due to changes in splicing of other than titin transcripts.  
